# Drug induced pancreatitis: A systematic review of case reports to determine potential drug associations

**Dianna Wolfe**[1], **Salmaan Kanji**[1,2], **Fatemeh Yazdi**[1], **Pauline Barbeau**[1], **Danielle Rice**[1], **Andrew Beck**[1], **Claire Butler**[1], **Leila Esmaeilisaraji**[1], **Becky Skidmore**[1], **David Moher**[1,3], **Brian Hutton**[1,3]*

**1** Clinical Epidemiology Program, Ottawa Hospital Research Institute, Ottawa, Ontario, Canada, **2** Department of Pharmacy, The Ottawa Hospital, Ottawa, Ontario, Canada, **3** School of Epidemiology, Public Health and Preventive Medicine, University of Ottawa, Ottawa, Ontario, Canada

* bhutton@ohri.ca

## Abstract

### Objective

A current assessment of case reports of possible drug-induced pancreatitis is needed. We systematically reviewed the case report literature to identify drugs with potential associations with acute pancreatitis and the burden of evidence supporting these associations.

### Methods

A protocol was developed a priori (PROSPERO CRD42017060473). We searched MEDLINE, Embase, the Cochrane Library, and additional sources to identify cases of drug-induced pancreatitis that met accepted diagnostic criteria of acute pancreatitis. Cases caused by multiple drugs or combination therapy were excluded. Established systematic review methods were used for screening and data extraction. A classification system for associated drugs was developed a priori based upon the number of cases, re-challenge, exclusion of non-drug causes of acute pancreatitis, and consistency of latency.

### Results

Seven-hundred and thirteen cases of potential drug-induced pancreatitis were identified, implicating 213 unique drugs. The evidence base was poor: exclusion of non-drug causes of acute pancreatitis was incomplete or poorly reported in all cases, 47% had at least one underlying condition predisposing to acute pancreatitis, and causality assessment was not conducted in 81%. Forty-five drugs (21%) were classified as having the highest level of evidence regarding their association with acute pancreatitis; causality was deemed to be probable or definite for 19 of these drugs (42%). Fifty-seven drugs (27%) had the lowest level of evidence regarding an association with acute pancreatitis, being implicated in single case reports, without exclusion of other causes of acute pancreatitis.

**Data Availability Statement:** All relevant data are within the manuscript and its Supporting Information files.

**Funding:** This work was supported by a team grant (awarded to authors DM and BH) from the Canadian Institutes of Health Research through the Drug Safety and Effectiveness Network (https://cihr-irsc.gc.ca/e/39389.html). The funders had no role in study design, data collection and analysis, decision to publish, or preparation of the manuscript.

**Competing interests:** We have read the journal's policy and the authors of this manuscript have the following competing interests: BH has previously received honoraria from Eversana (previously Cornerstone Research Group) for provision of methodologic advice related to the conduct of systematic reviews and meta-analysis. This does not alter our adherence to PLOS ONE policies on sharing data and materials. All other authors have no conflicts of interest to disclose.

## Discussion

Much of the case report evidence upon which drug-induced pancreatitis associations are based is tenuous. A greater emphasis on exclusion of all non-drug causes of acute pancreatitis and on quality reporting would improve the evidence base. It should be recognized that reviews of case reports, are valuable scoping tools but have limited strength to establish drug-induced pancreatitis associations.

## Registration

CRD42017060473.

## Introduction

Acute pancreatitis (AP) is a common gastrointestinal cause of hospitalization, with over 230,000 cases per year leading to hospitalization in the United States [1]. Drug-related causes of AP are rare (0.1–2% of cases) [2], but occasionally can be life threatening. Knowledge of drugs with the potential to cause AP may aid in clinician awareness of this uncommon etiology, resulting in prevented re-administration of the offending medication and avoided patient harm.

Several publications have listed the drugs more commonly associated with AP [3–5,2,6,7]. However, a comprehensive review of the literature to identify drugs with potential associations has not been conducted since 2006 [7]. We aimed to update the previous systematic review of case reports by Badalov et al. [7] to develop a current list of drugs with potential associations with AP. However, early in our work we identified several limitations in the reporting of that review that prevented the performance of a formal review update. Consequently, a full systematic review of the case report literature from inception was performed to identify all rigorously diagnosed cases of AP that were suspected cases of drug-induced pancreatitis (DIP). With these data, we sought to classify the suspected drugs according to the level of evidence available upon which a potential association could be based.

## Methods

A systematic review protocol was developed a priori and registered with PROSPERO (CRD42017060473). Our approach to data synthesis deviated from the protocol after we realized that the drug classification system used by Badalov et al. [7] was data driven and specific to the previous review, and that a more rigorous and global classification system was needed.

### Research question addressed

This review addressed the following primary research question: "*What drugs have potential associations with drug-induced pancreatitis? What level of evidence is available for these associations?*"

### Study eligibility criteria

The population-intervention-comparator-outcomes-study design (PICOS) framework was used to identify eligible cases. Details of the criteria established a priori were as follows:

- *Population*. Only human patients, with no restrictions on age or other demographics.

- **Intervention and comparator**. Any drug that was suspected of causing acute pancreatitis in the reported case was of interest. Drug combinations were excluded (e.g., combination HIV therapy, cancer chemotherapy regimens). An exception was made for trimethoprim-sulfa-methoxazole as these drugs are routinely administered in the same relative proportions. Where multiple drugs were administered prior to AP, we included cases in which the authors implicated an individual drug and excluded cases in which a combination of drugs or their interaction were suspected of causing the AP. We excluded herbal products, vaccines, poisons, and insecticides. Based on the criteria used by Badalov et al. [7], we limited inclusion to those cases that reported the name and dosage of the drug thought to cause the AP. No comparator was required.

- **Outcomes**. We deviated from the review by Badalov et al [7] in that we included only cases that diagnosed acute pancreatitis (AP) using accepted criteria [8]. These criteria were initially published in 2006, the year in which Badalov et al. conducted their review. At least two of the following three features were required to be present for a diagnosis of AP to be upheld and a case included:

  ○ Typical clinical symptoms (e.g., epigastric pain, nausea, vomiting);

  ○ Serum amylase or lipase elevated at least three times the upper limit of normal (ULN). When the ULN was not reported, we used an ULN of 160 units/l for both amylase and lipase [9] (i.e., the reported amylase and/or lipase must have been >480 units/l for the criteria of serum amylase or lipase > three times the ULN to be accepted);

  ○ Characteristic imaging findings of AP on contrast-enhanced computed tomography (CECT), on magnetic resonance imaging (MRI) or transabdominal ultrasonography.

  The etiologic cause of the AP must have been reported as being a drug exposure, with drug exposure occurring prior to the development of signs of AP. The time between initiation of drug administration and occurrence of AP (the latency) must have been reported. We did not discriminate according to the efforts that were undertaken to determine causality or the likelihood of the association. Cases suspected of having chronic pancreatitis with a drug-induced exacerbation were excluded.

- **Study design**. We included only case reports and case series published in full text. Reviews of cases previously published in the literature were excluded to avoid case duplication. Letters to the editor were included, if all other criteria for inclusion were satisfied.

### Searching the literature

An experienced medical information specialist (BS) developed and tested the search strategy using an iterative process in consultation with the review team. Another senior information specialist peer reviewed the strategy prior to execution using the PRESS Checklist [10]. We conducted several systematic reviews related to drug-induced pancreatitis concurrently and utilized the same base strategy for all. We performed separate searches for primary studies (for the concurrent systematic reviews) and case reports. Using the Ovid platform, we searched Ovid MEDLINE®, including Epub Ahead of Print and In-Process & Other Non-Indexed Citations, and Embase Classic+Embase. The Cochrane Library on Wiley, was also searched for primary studies alone. All database searches were performed on 31 January 2017, with a search update on 28 March 2019. We undertook a grey literature search of clinical practice guideline registries, the TRIP database, and Google Scholar on 9–10 March 2017.

We incorporated controlled vocabulary (e.g., "Pancreatitis/ci [chemically induced]", "Drug-Related Side Effects and Adverse Reactions") and keywords (e.g., "drug-induced pancreatitis", "adverse effect", "detection") into the searches. We applied research design filters for both primary studies and case reports. Vocabulary and syntax were adjusted across the databases. No date or language limits were applied to either of the searches, but we removed animal-only and opinion pieces, where possible, from the results (opinion pieces retained for the case report search).

Specific details regarding the strategies are provided in **S1 Text**. References identified by the case report search were de-duplicated against the references identified in the primary study search to reduce reference screening load.

## Process of study selection

A two-stage study selection process was used: all titles and abstracts were initially screened for potential relevance, with full texts of the potentially relevant references being screened subsequently. Screening at both stages was conducted independently by two reviewers, with included references requiring assessment by only one reviewer at Stage 1, but agreement of both reviewers required for inclusion at Stage 2. Agreement of both reviewers was required for exclusion at both stages. The online systematic review software DistillerSR (Evidence Partners Inc., Ottawa, Canada) was used to operationalize screening, using forms developed by the review team that were piloted prior to both stages of screening to maximize reviewer agreement. Conflicts were resolved by consensus, with consultation with a third independent reviewer if necessary.

## Data extraction and risk of bias assessment

Data were extracted in Microsoft Excel (Microsoft Corp, Seattle, WA), using an extraction template that was initially piloted by two users on a set of five randomly selected case reports and adjusted as necessary. Five reviewers extracted data, and each conducted a pilot to improve agreement between extractors. Data were extracted by one reviewer and verified by a second reviewer.

Data elements collected during the extraction process included the following:

- Author, year of publication, and country of case;

- Patients' key characteristics, including age, sex, underlying comorbidities, presence of renal dysfunction or other risk factors;

- Reported drug of association, including dosage and/or whether the case was an overdose related to a suicide attempt;

- Reported latency (i.e., the time from first drug exposure to the onset of symptoms of pancreatitis);

- Criteria for AP diagnosis, including presence/absence of typical symptoms, elevated amylase/lipase levels, and abnormalities on imaging, laparotomy, or autopsy;

- Exclusion of other causes of AP, including alcohol, biliary causes, hypertriglyderidemia/ hyperlipidemia, hypercalcemia, autoimmune causes, genetic causes, anatomic causes, trauma, viral causes, other drugs, and other causes;

- Presence/absence of a formal causality assessment; and

- Presence/absence of a re-challenge.

For data elements related to exclusion of other causes of AP, the extracted data were constrained to a set of predefined response options (i.e., "Yes," "No," "Unclear," and blank), using the data validation tool in Excel. Constraint of response options at the data collection phase was considered necessary to simplify drug classification at the data synthesis phase, given the high number of included case reports expected. "Yes" was was selected to indicate that a specific cause had been worked up and excluded. "No" was selected if a cause had been worked up and could not be excluded (e.g., hypertriglyceridemia at presentation that may have been present prior to drug exposure). Where a cause was worked up and the findings were not clearly interpretable, "Unclear" was selected. A blank response indicated the cause had not been reported in the publication. For each case report, the decision regarding whether a specific cause was excluded or not was based mainly upon the case report authors' interpretation and their reported normal ranges (e.g., hypertriglyceridemia, hyperlipidemia, hypercalcaemia). If normal ranges were not provided, we referred to published sources [9]. We did not infer whether the non-drug causes had been worked up sufficiently to current standards to allow exclusion (e.g., whether currently accepted testing had been conducted to eliminate biliary causes, whether all relevant viruses had been eliminated with appropriate testing). Some non-drug causes required definition. Alcohol as a cause of AP was considered only to be chronic alcohol abuse or acute alcohol toxicity, as defined by the case report author, and not any alcohol use. Alcoholism as a cause was assumed to be excluded in patients under 18 years of age. A positive result for any autoimmune, genetic, or viral test elicited a "No" response for exclusion of that cause. However, when all autoimmune, genetic, or viral testing was negative, a "Yes" response was selected, regardless if all testing for that cause had been conducted. Any mention of family history of pancreatitis or hyperlipidemia, whether negative or positive, elicited a "Yes" or "No" response, respectively, regarding exclusion. Anatomic causes (e.g., irregularities of the pancreatic duct, pancreas divisum) were considered excluded if specifically mentioned as absent in imaging findings.

We critically scrutinized each case report in detail to make inferences as to whether other drugs were sufficiently excluded as a potential cause of the AP. We considered other drugs to have been excluded if (1) an effort had been made to report that other drugs were not associated (e.g., the authors stated that no AP case reports associated with the other drug(s) had been published to date); (2) the patient wasn't taking other drugs; (3) the patient was taking other drugs but they were continued/restarted without a recurrence of AP; or (4) a formal causality assessment was conducted specific to the drug of interest (e.g., using Naranjo criteria [11]) that indicated that the drug of interest was the probable or definite cause and other drugs assessed with the same criteria were not. Other drugs were not considered to have been excluded if (1) the authors stated that they couldn't exclude other drugs or (2) treatment with other drugs was temporally associated with an initial AP episode and AP recurred, while the patient was taking both the drug of interest and the other drug(s), with the authors not providing clear reasoning as to why the other drugs should be excluded. An "Unclear" response was warranted for all other cases.

Underlying diseases or conditions that may predispose a patient to AP were extracted, including the following: inflammatory bowel disease (i.e., Crohn's disease and ulcerative colitis), diabetes mellitus, liver disease (e.g., any viral hepatitis, cirrhosis, cholangitis, liver transplant, Alagille's Syndrome, signs of ascites and/or jaundice), HIV/AIDS, dyslipidemia (e.g., hypercholesterolemia, hypertriglyceridemia), immunologic disorders (e.g., rheumatoid arthritis, systemic lupus erythematosus, glomerulonephritis, bullous pemphigoid, autoimmune pancreatitis, optic neuritis), and renal dysfunction that could potentially lead to reduced drug clearance (e.g., end-stage renal disease, acute or chronic renal disease/insufficiency/failure, glomerulonephritis, lupus nephritis, IgA nephropathy, nephrotic syndrome, renal transplant

+/- rejection, renal carcinoma, nephrolithiasis, prior drug nephrotoxicity, dialysis treatment, membranous glomerulopathy, metastatic involvement of kidney, nephrectomy, pyelonephritis, other renal insufficiency). The presence of an overdose, whether intentional or unintentional, was also extracted, although we did not compare the dose of the administered drug to current label dosages.

Data pertaining to formal causality assessment were extracted, including the name of the tool used and the outcome of the assessment. Following data extraction, case reports were manually screened for duplicates by sorting on patient and publication characteristics to ensure duplication was not present in our final database.

A tool has recently been proposed to assess the methodological quality of case reports and case series included in systematic reviews [12]. This tool proposes broad explanatory questions similar to the detailed criteria that we used either during study selection (e.g., only cases using accepted criteria for AP diagnosis and with complete reporting of drug dosage and latency were included) or to assess DIP causality in our included cases. Given that the questions in the tool considered most critical to assessment of methodological quality in our review context had already been assessed and accounted for, we elected not to conduct a separate risk of bias evaluation.

## Summarizing the evidence

Descriptive statistics of publication characteristics and patient demographic variables were estimated. Case report data were grouped by the drug suspected to have caused the case of AP. Drugs were grouped, based on discussion with our clinical experts. Drugs were not separated by route of administration (e.g., oral and aerosolized pentamidine were considered the same drug). Oral contraceptives were grouped together, as the component combinations may be numerous and the mechanism of causing AP was likely the same. However, other single-component estrogen-like therapies were considered as separate medications (e.g., diethylstilbestrol). Asparaginase medications were grouped as some case reports did not clarify the type of asparaginase administered (e.g., obtained from *Erwinia chrysanthemi* versus *Escherichia coli*, pegylated versus non-pegylated asparaginase). Cases involving interferon alpha and beta were grouped separately, as were all corticosteroids (e.g., prednisone, prednisolone, dexamethasone, etc.).

Within each drug grouping, case reports were assessed according to the classification criteria described in Table 1. No validated classification criteria exist that assess the level of evidence of an association of a drug with AP. Our criteria were developed a priori by the review team, loosely based upon the data-driven classification system reported by Badalov et al. [7]. Because the Badalov system was derived from their review data and not developed a priori, there were classification gaps in which some drugs could fall if the system were to be used on a different set of data. The classification system used in the current review was structured to close these gaps. Additionally, our system has limited the impact of publication bias in the reporting of case reports (e.g., reporting of potential cases of DIP for some drugs may increase based upon perceptions of potential associations in the medical community). Drugs having multiple case reports associated with them are recognized as having a greater potential association than those with single case reports. However, beyond this, there is no arbitrary number of case reports required to increase the level of evidence of an association. Ultimately, we defined six drug classes based upon (1) evidence of a positive re-challenge, (2) a simplified measure of the rigour of the causality assessment conducted (i.e., whether three other main causes of AP —alcohol, biliary, and hypertriglyceridemia/hyperlipidemia—and all other drugs were conclusively ruled out as causes of DIP), and (3) the consistency of the latency for drugs for which

**Table 1. Drug classification system for assessment of association with DIP.**

| Drug class | Definition |
|---|---|
| Class Ia | • At least 1 case report in humans, with positive re-challenge |
| | • All other causes, such as alcohol, hypertriglyceridemia (and hyperlipidemia), gallstones, and other drugs are ruled out |
| Class Ib | • At least 1 case report in humans, with positive re-challenge |
| | • Other causes, such as alcohol, hypertriglyceridemia, gallstones, and other drugs were not ruled out |
| Class Ic | • At least 1 case report in humans, without a positive re-challenge (i.e., no re-challenge or a negative re-challenge) |
| | • Other causes, such as alcohol, hypertriglyceridemia, gallstones, and other drugs are ruled out |
| Class II | • At least 2 cases in humans reported in the literature, without a positive re-challenge (i.e., no re-challenge or a negative re-challenge) |
| | • Other causes, such as alcohol, hypertriglyceridemia, gallstones, and other drugs were not ruled out |
| | • Consistent latency* |
| Class III | • At least 2 cases in humans reported the literature, without a positive re-challenge (i.e., no re-challenge or a negative re-challenge) |
| | • Other causes, such as alcohol, hypertriglyceridemia, gallstones, and other drugs were not ruled out |
| | • Inconsistent latency* |
| Class IV | • At least 1 case in humans reported the literature |
| | • Drugs not fitting into the earlier-described classes |

* "Consistent latency" defined as >75% of cases falling into the same latency category

• Category 1: <24h.

• Category 2: 1–30 days.

• Category 3: >30 days.

cases with a either a positive rechallenge or a rigorous causality assessment were not reported. We used latency categories based upon those presented by Badalov et al. [7] (e.g., <24 hours, 1–30 days, >30 days), and have defined "consistent latency" as >75% of case reports for a drug falling into the same latency category. We conducted sensitivity analyses to evaluate (1) the impact of requiring positive imaging findings in the diagnosis of AP, and (2) the impact of requiring all ten causes of AP to be excluded instead of only four for a diagnosis of DIP in classes Ia and Ic. Classification of drugs was aided by the filtering tool in Excel (Microsoft Corp., Seattle, WA).

Latency data were categorized and the predominant latency category was identified for each drug (i.e., the latency category in which the majority of cases for each drug fell). The latency consistency of a drug was defined as the proportion of its cases that fell in its predominant latency category. Drugs represented by two cases that fell in two different latency categories were assigned a consistency of 0%.

## Reporting of review findings

The reporting in this manuscript was guided by the PRISMA Statement [13]. A completed PRISMA Checklist documents the completeness of reporting (see S2 Text).

## Results

A total of 596 publications were included, encompassing 713 unique cases and 213 unique drugs (Fig 1); detailed listings of included studies and excluded studies are provided in S3 Text and S4 Text, respectively. The characteristics of the included case reports and the

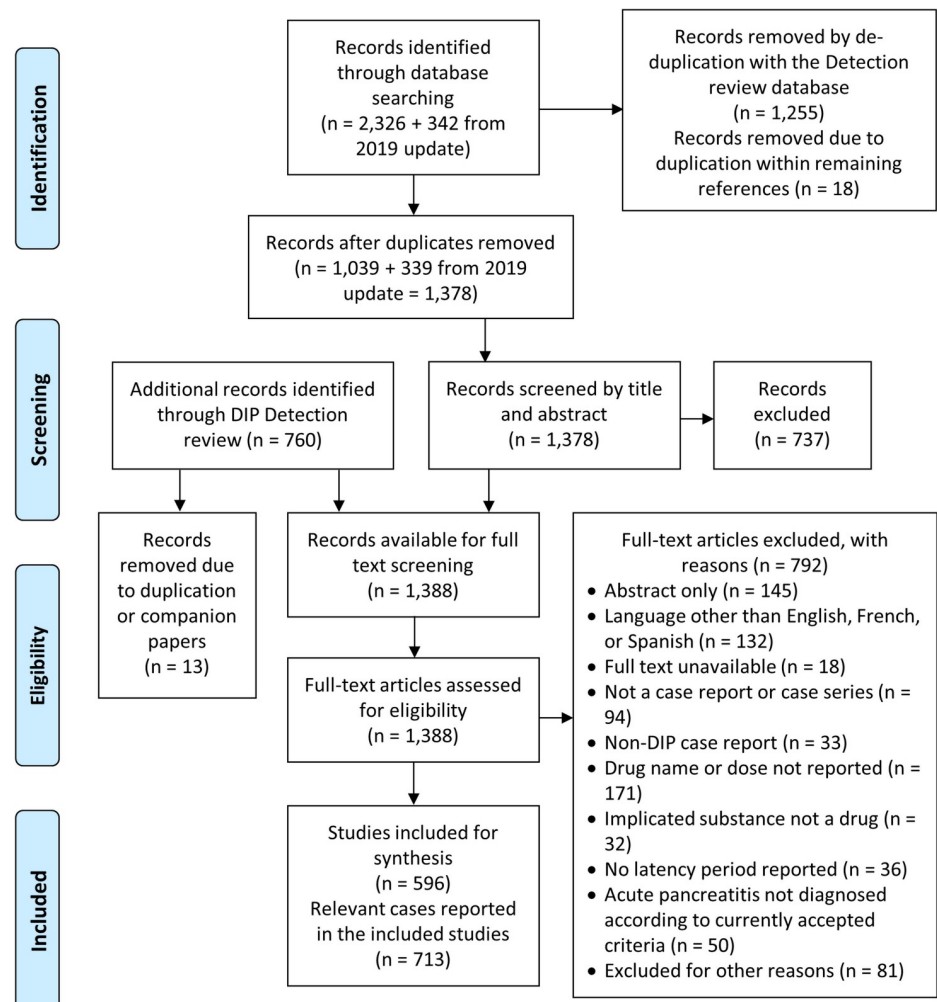

**Fig 1. Flow diagram of the study selection process.**

demographics of the included patients are presented in Table 2 and Table 3, respectively. The case report publication rate was highest from 1990 to 1999 (n = 241, 24.1 cases/year), and has slowed in recent years (2010–early 2019: 19.1 cases/year). In most case reports, the causes of AP that were excluded to arrive at a diagnosis of DIP were poorly reported, with only two recent case reports having data reported for all ten non-drug causes for which we extracted data [14,15]. Similarly, 81% of cases did not conduct a formal causality assessment. Overall, 182 of 190 re-challenges that were conducted resulted in a recurrence of AP (96%). However, re-challenge was not performed for most cases, with many citing the ethical concerns of knowingly re-administering a drug with the potential to cause a life-threatening condition.

Half of all cases (n = 358; 50%) had at least one primary underlying condition or risk factor for pancreatitis listed in Table 3. Thirty-nine percent (n = 275) had at least one primary underlying condition identified as necessary to rule out before a drug etiology should be considered. In addition, 53 patients (7%) had renal insufficiency and one (0.1%) had hepatosteatosis, which may alter drug clearance or metabolism, respectively, increasing the risk of adverse drug reactions; 47 patients (7%) had had a cholecystectomy, which may influence exocrine pancreatic dynamics; and 6 patients (0.8%) had had a previous episode of pancreatitis. Seven

**Table 2. Characteristics of the included case reports (n = 713).**

| Characteristic | Cases (%) |
| --- | --- |
| **Year of publication** | |
| 1960–69 | 6 (0.8) |
| 1970–79 | 21 (2.9) |
| 1980–89 | 80 (11.2) |
| 1990–99 | 241 (33.8) |
| 2000–2009 | 193 (27.1) |
| 2010–2019 | 172 (24.1) |
| **Country** | |
| USA | 212 (29.7) |
| France | 85 (11.9) |
| Spain | 72 (10.1) |
| The Netherlands | 38 (5.3) |
| Japan | 34 (4.8) |
| UK | 29 (4.1) |
| Italy | 25 (3.5) |
| India | 24 (3.4) |
| Canada | 22 (3.1) |
| Other | 172 (24.1) |
| **Causes assessed as excluded to arrive at diagnosis of DIP** | |
| Alcoholism | Excluded: 547 (76.7) |
| | Not excluded: 14 (2.0) |
| | Unclear: 12 (1.7) |
| | Not reported: 140 (19.6) |
| Gallstones/ biliary disease | Excluded: 494 (69.2) |
| | Not excluded: 32 (4.5) |
| | Unclear: 11: (1.5) |
| | Not reported: 176 (24.7) |
| Hyperlipidemia/ hypertriglyceridemia | Excluded: 304 (42.6) |
| | Not excluded: 58 (8.1) |
| | Unclear: 7 (1.0) |
| | Not reported: 344 (48.2) |
| Hypercalcemia | Excluded: 285 (40.0) |
| | Not excluded: 12 (1.7) |
| | Unclear: 2 (0.3) |
| | Not reported: 414 (58.1) |
| Autoimmune disease | Excluded: 51 (7.2) |
| | Not excluded: 4 (0.6) |
| | Unclear: 5 (0.7) |
| | Not reported: 653 (91.6) |
| Genetic causes or family history of AP | Excluded: 58 (8.1) |
| | Not excluded: 4 (0.6) |
| | Unclear: 2 (0.3) |
| | Not reported: 649 (91.0) |
| Anatomic | Excluded: 54 (7.6) |
| | Not excluded: 6 (0.8) |
| | Unclear: 1 (0.1) |
| | Not reported: 652 (91.4) |

(*Continued*)

**Table 2.** (Continued)

| Characteristic | Cases (%) |
|---|---|
| Trauma | Excluded: 113 (15.8) |
| | Not excluded: 1 (0.1) |
| | Unclear: 1 (0.1) |
| | Not reported: 598 (83.9) |
| Viral | Excluded: 166 (23.2) |
| | Not excluded: 16 (2.2) |
| | Unclear: 4 (0.6) |
| | Not reported: 527 (73.9) |
| Other drugs | Excluded: 492 (69.0)[a] |
| | Not excluded: 97 (13.6)[b] |
| | Unclear: 120 (16.8)[c] |
| | Not reported: 0 (0)[d] |
| **Formal causality assessment conducted** | |
| Yes | 137 (19.2) |
| No | 576 (80.8) |
| **Causality assessment tools used in case reports that assessed causality (n = 137)** | |
| Naranjo criteria[e] | 68 (49.6) |
| Eland[f] | 24 (17.5) |
| Delcenserie 2001 | 14 (10.2) |
| Mallory and Kern | 11 (8.0) |
| Karch and Lasagna | 10 (7.3) |
| Other French assessment tools[g] | 8 (5.8) |
| Other tools[h] | 6 (4.4) |
| **Causality assessment findings in case reports that assessed causality (n = 137)** | |
| Definite/highly probable | 17 (12.4) |
| Probable/likely | 100 (73.0) |
| Probable/likely or possible | 2 (1.5) |
| Possible/plausible | 16 (11.7) |
| Doubtful | 2 (1.5) |
| **Re-challenge conducted** | |
| Yes, and positive | 182 (25.5) |
| Yes, and negative | 8 (1.1) |
| No | 523 (73.4) |

[a] "Excluded" indicates that an effort was made by the authors to report that other drugs were not associated OR the patient wasn't taking other drugs OR the patient was taking other drugs but they were continued/restarted without a recurrence of AP OR causality was assessed as "probable" for drug of interest.

[b] "Not excluded" indicates that the author stated that they couldn't exclude other drugs OR the patient started taking other drugs at the time that the pancreatitis started.

[c] "Unclear" indicates that the patient was taking other drugs for a period prior to the AP and no comment was made regarding their possible association OR causality was assessed as "possible" for drug of interest.

[d] "Not reported" could not apply to any case because cases in which the impact of other drugs was not reported were categorized as "Unclear".

[e] Three cases reported assessment findings for both Naranjo and WHO-UMC tools.

[f] All cases reporting using the Eland algorithm were published in the paper by Eland that described the causality assessment tool.

[g] Included Begaud et al (1985) (n = 6), Dangoumau et al. (1978) (n = 1), and Delcenserie et al. (1992) (n = 1).

[h] Included WHO-UMC assessment tool (n = 4), FDA algorithm (n = 1), and Kramer's algorithm (n = 1).

**Table 3. Patient demographics of the included case reports (n = 713).**

| Demographic | Cases (%) |
|---|---|
| **Patient age (years)** | |
| <1 | 1 (0.1) |
| 1–12 | 66 (9.3) |
| 13–18 | 62 (8.7) |
| 19–24 | 65 (9.1) |
| 25–64 | 396 (55.5) |
| >64 | 122 (17.1) |
| Not reported | 2 (0.3) |
| **Sex** | |
| Female | 339 (47.5) |
| Male | 373 (52.3) |
| Not reported | 1 (0.1) |
| **Primary underlying conditions that may predispose to AP[a]** **(one per patient, if present; n = 275)** | |
| Crohn's disease/inflammatory bowel disease/ulcerative colitis | 78 (10.9) |
| Diabetes | 61 (8.6) |
| Genetic disorder (Class V Cystic Fibrosis mutation) | 2 (0.3) |
| Hepatitis | 27 (3.8) |
| HIV/AIDS[b] | 47 (6.6) |
| Hyperlipidemia/hypercholesterolemia/hypertriglyceridemia | 33 (4.6) |
| Immune disorder[c] | 26 (3.6) |
| Infection (malaria) | 1 (0.1) |
| **Other potential risk factors for AP or DIP** **(multiple per patient possible)** | **Cases (%)** |
| Previous cholecystectomy | 47 (6.6) |
| Previous episode of pancreatitis[d] | 6 (0.8) |
| Possible renal dysfunction[e] | 53 (7.4) |
| Hepatic disease[f] | 13 (1.8) |
| Gall stones/ biliary disease[g] | 32 (4.5) |
| History of moderate-to-heavy alcohol use or abuse | 14 (2.0) |

[a] Primary underlying conditions that were identified by our content experts as necessary to rule out before a diagnosis of DIP should be considered.

[b] Patients with HIV/AIDS often had other underlying infections.

[c] Includes rheumatoid arthritis, systemic lupus erythematosus, autoimmune skin disorders and glomerulonephritis, psoriatic arthritis, optic neuritis.

[d] Includes autoimmune, drug-induced, and gallstone-induced pancreatitis, as well as pancreatic carcinoma, etc.

[e] Includes acute/chronic renal insufficiency, end-stage renal disease, pyelonephritis, glomerulonephritis, glomerulopathy, nephrotic syndrome, nephrolithiasis, nephropathy, renal carcinoma, metastatic cancer, renal transplant rejection, etc. Renal function did not have to be tested for case to be flagged as possible renal dysfunction.

[f] Either hepatitis in addition to one of the above primary conditions or other hepatopathy, including HBV, HCV, previous biliary problems not treated by cholecystectomy, chronic active hepatitis, hepatocellular carcinoma, primary sclerosing cholangitis, and hepatosteatosis.

[g] Biliary disease was present or could not be ruled out at time of AP diagnosis.

cholecystectomized patients had an opioid or opioid receptor agonist implicated (15%), four oral hypoglycemic agent implicated, and three a tetracycline derivative implicated. Four percent of DIP cases (n = 29) were due to intentional or unintentional overdose, with another

three cases being unclear as to whether there was an overdose. Ninety-eight cases (14%) did not fully report ruling out all other drugs as potential causes, calling into question the drug ultimately identified as the causative agent.

The 213 unique drugs implicated in the case reports were categorized according to our a priori classification scheme (Table 4). Similar numbers of drugs met the criteria of Classes Ia, Ib, and Ic (n = 45, 46, and 54, respectively). Most drugs with multiple case reports were categorized in these upper classes, with few remaining to be categorized as Class II (n = 6) or III (n = 5). More than a quarter of drugs were found in single case reports that did not have either exclusion of other causes of AP or a positive rechallenge and were categorized as Class IV (n = 57; 27%).

Negative re-challenge (i.e., no recurrence of AP upon re-administration of the drug) occurred in eight cases representing eight different drugs—all-trans retinoic acid, brentuximab vedotin, clozapine, interferon alpha, L-asparaginase, sorafenib, tacrolimus, and valproic acid—in different drug classes. All but brentuximab vedotin had been implicated in other case reports. Two of the eight cases with negative re-challenges were identified to have a "probable" association with the reported drug on using either the Naranjo criteria [11] or criteria of Delcenseri et al. [610] (brentuximab vedotin and L-asparaginase).

A drug-level summary of the identified case reports is provided in the review supplement (S5 Text), as is a comprehensive summary of the latency data associated with all agents and number of cases meeting the class criteria for eachd drug. A summary of key information for each of the drug classes (as outlined in Table 1) is provided in the following sections.

## Identified Class Ia drugs

Forty-five drugs were classified as Class Ia from 344 case reports (48%), for a median of 4 case reports per drug (mean = 7.6; mode = 1; range = 1–56). Twelve drugs were included in Class Ia based on one case report each. The three drugs reported in the highest number of cases were valproic acid (n = 56), L-asparaginase (n = 34), and 5-ASA (n = 31). Predominant latency categories, in order of frequency of occurrence amongst the 45 drugs, were 1–30 days (n = 27 drugs), >30 days (n = 14 drugs), and <24 hours (n = 4 drugs). The median latency consistency amongst the 34 drugs with multiple case reports was 75%.

Causality was assessed for 25 drugs (56% of Class Ia drugs) in 64 cases (19% of Class Ia cases). Four drugs demonstrated less than probable likelihoods of causation that were assessed in single case reports: acetaminophen (likely), captopril (possible), tetracycline (possible), and valproic acid (plausible). All other drugs for which causality was assessed demonstrated probable or definite likelihood in at least one case report.

Forty-three percent of the 344 cases (n = 147) had an underlying condition that may have predisposed them to AP. Twenty-three cases (7%) had possible renal dysfunction that may have contributed to poor drug clearance and a predisposition to a drug reaction. Thirteen cases had both an underlying condition and renal dysfunction potentially predisposing them to both AP and a drug reaction.

## Identified Class Ib drugs

Forty-six drugs were classified as Class Ib from 175 case reports (26%), for a median of 2 case reports per drug (mean = 3.9; mode = 1; range = 1–20). The three drugs reported in the highest number of cases were pentamidine (n = 20), stibogluconate (n = 18), and propofol (n = 12). Predominant latency categories, in order of frequency of occurrence amongst the 46 drugs, were 1–30 days (n = 30 drugs), >30 days (n = 14 drugs), and <24 hours (n = 2 drugs). The median latency consistency for the 31 drugs with multiple case reports was 80%.

**Table 4. Drugs associated with DIP in the included case reports (n = 713).**

| Class Ia (n = 45 drugs) | Class Ib (n = 46 drugs) | Class Ic (n = 53 drugs) | Class II (n = 6 drugs) | Class III (n = 5 drugs) | Class IV (n = 57 drugs) |
|---|---|---|---|---|---|
| 5-acetylsalicylic acid (mesalamine); 31 cases [16–38] | Amiodarone; 3 cases [39–41] | **Adefovir dipivoxil; 1 case [42] | Ceftriaxone; 5 cases [43–47] | **Acetylsalicylic acid; 3 cases [48,49] | **Ado-trastuzumab emtansine; 1 case [50] |
| 6-mercaptopurine (6-MP); 6 [51–55] | Ampicillin; 1 [56] | **Amoxicillin + clavulanic acid; 2 [14,57] | **Clofibrate; 2 [58] | Gold; 4 [59–61] | **Albiglutide; 1 [62] |
| Acetaminophen; 9 [63–70] | **Antilymphocyte globulin; 1 [71] | **Artesunate 1 [72] | **Exenatide; 3 [73–75] | **Nivolumab; 2 [76,77] | Alendronate; 1 [25] |
| All-trans retinoic acid; 3 [78–80] | Carbamazepine; 5 [81–85] | Atorvastatin; 3 [86–88] | Isotretinoin; 3 [89–91] | **Ondansetron; 2 [92,93] | **Amineptine; 1 [94] |
| Azathioprine; 29 [25,30,95–112] | **Ciprofloxacin; 1 [113] | **Axitinib; 2 [114,115] | **Levetiracetam; 2 [116,117] | Tacrolimus; 2 [118,119] | Benazepril; 1 [120] |
| Azodisalicylate/ olsalazine; 3 [25,121,122] | Clomiphene; 2 [123,124] | **Boceprevir; 1 [125] | **Sitagliptin; 3 [126–128] | | **Brentuximab vedotin; 1 [129] |
| Bezafibrate; 1 [130] | **Clothiapine; 1 [131] | **Bortezomib; 2 [132,133] | | | **Calcium carbonate; 1 [134] |
| Captopril; 3 [25,135,136] | Clozapine; 9 [137–145] | **Canaglifozin; 2 [146,147] | | | Capecitabine; 1 [148] |
| Carbimazole; 2 [149,150] | **Cytarabine; 4 [151–153] | **Candesartan; 1 [154] | | | Chlorthalidone; 1 [155] |
| Cimetidine; 7 [25,156–160] | Dexamethasone; 1 [161] | **Celecoxib; 4 [162–165] | | | **Ciprofibrate; 1 [25] |
| Codeine; 5 [166,167] | Didanosine; 8 [25,168–172] | Clarithromycin; 5 [173–177] | | | Cisplatin; 1 [178] |
| Dapsone; 3 [179–181] | **Diphenoxylate + atropine; 1 [182] | Danazol; 1 [183] | | | **Clomipramine; 1 [184] |
| Erythromycin; 12 [185–196] | **Eluxadoline 2 [197,198] | **Dexfenfluramine; 1 [199] | | | **Clonidine; 1 [200] |
| Fluvastatin; 1 [201] | Enalapril; 10 [25,202–208] | Diclofenac; 2 [209,210] | | | **Demeclocycline; 1 [211] |
| Furosemide; 4 [212–215] | **Everolimus; 2 [216,217] | **Diethylstilbestrol; 1 [218] | | | **Doxylamine succinate; 1 [219] |
| Interferon-alpha; 12 [220–230] | **Growth Hormone; 2 [231,232] | **Dilantin; 1 [233] | | | **Ertapenem; 1 [234] |
| Isoniazid; 10 [235–244] | Hydrochlorothiazide; 2 [245,246] | **Dimethyl fumarate; 1 [247] | | | **Estramustine phosphate; 1 [248] |
| L-asparaginase; 34 [249–269] | Hydrocortisone; 1 [270] | **Doxycycline; 4 [25,271–273] | | | Famcyclovir; 1 [274] |
| Lisinopril; 6 [275–280] | Ifosfamide; 4 [281–284] | **Ezetimibe; 1 [285] | | | **Gatifloxacin; 1 [286] |
| Metformin; 4 [287–290] | **Indalpine; 1 [291] | Finasteride; 1 [292] | | | Gemfibrozil; 1 [293] |
| Methimazole; 6 [294–299] | Lamivudine; 3 [25,300,301] | **Flurbiprofen; 1 [302] | | | **Granisetron; 1 [303] |
| **Methylprednisolone; 6 [304–309] | Losartan; 3 [310–312] | **Gadolinium; 3 [313–315] | | | Interleukin-2; 1 [316] |
| Metronidazole; 11 [25,317–326] | **Mefenamic acid; 3 [63,327,328] | **Glicazide; 1 [329] | | | **Lacosamide; 1 [330] |
| Nitrofurantoin; 3 [331–333] | Meglumine antimoniate; 5 [334–337] | **Glimepiride; 1 [338] | | | Lamotrigine; 1 [339] |
| **Orlistat; 1 [340] | Methyldopa; 3 [63,341] | **Ibuprofen; 3 [25,342,343] | | | **Linagliptin 1 [344] |
| **Piroxicam; 1 [345] | Mirtazapine; 4 [346–349] | Indomethacin; 2 [350,351] | | | **Linezolid 1 [352] |
| Pravastatin; 2 [353,354] | Nelfinavir; 1 [355] | Interferon beta; 1 [356] | | | **Lixisenatide 1 [357] |
| Prednisone; 8 [358–363] | Octreotide; 6 [364–368] | Irbesartan; 1 [369] | | | **Loperamide; 1 [370] |
| Premarin; 2 [371,372] | Omeprazole; 1 [373] | **Itraconazole; 2 [374] | | | Lovastatin; 1 [375] |

*(Continued)*

**Table 4.** (Continued)

| Class Ia (n = 45 drugs) | Class Ib (n = 46 drugs) | Class Ic (n = 53 drugs) | Class II (n = 6 drugs) | Class III (n = 5 drugs) | Class IV (n = 57 drugs) |
|---|---|---|---|---|---|
| Procainamide; 1 [376] | Oral contraceptive; 3 [377,378] | **Ixazomib; 1 [379] | | | **Maprotiline; 1 [25] |
| Pyritinol; 1 [380] | Oxyphenbutazone; 2 [25,381] | Ketoprofen; 2 [382,383] | | | **Methandrostenolone; 1 [384] |
| Ramipril; 4 [206,385–387] | Paclitaxel; 4 [388–391] | Ketorolac; 2 [392,393] | | | **Micafungin; 1 [394] |
| Ranitidine; 1 [395] | **Paromomycin; 1 [396] | **Lanreotide; 2 [397,398] | | | **Miltefosine; 1 [399] |
| Rosuvastatin; 1 [400] | Pentamidine; 20 [401–417] | **Lenvatinib; 1 [418] | | | ***Mizoribine; 1 [419] |
| Simvastatin; 6 [420–424] | **Perindopril; 2 [425,426] | **Liraglutide; 5 [427–431] | | | **Montelukast; 1 [432] |
| **Sorafenib; 7 [433–439] | Prednisolone; 4 [440,441] | **Meprobamate; 1 [442] | | | **Mycophenolate mofetil 1 [443] |
| Sulindac; 9 [25,444–450] | Propofol; 12 [15,451–460] | Metolazone; 2 [461,462] | | | **Nifuroxazide; 1 [463] |
| Tamoxifen; 6 [464–469] | **Quetiapine; 2 [470,471] | Minocycline; 5 [472–474] | | | **Norfloxacin; 1 [475] |
| **Telaprevir; 1 [476] | **Rifampicin; 1 [477] | **Naltrexone; 2 [478,479] | | | **Pazopanib 1 [480] |
| Tetracycline; 5 [25,481–483] | Risperidone; 4 [484–487] | Naproxen; 3 [488–490] | | | **Phenformin; 1 [491] |
| **Tigecycline; 10 [492–500] | **Salazopyrine; 1 [501] | **Nilotinib; 3 [502,503] | | | Phenolpthalien; 1 [504] |
| **Thalidomide; 1 [505] | **Saxagliptin; 1 [506] | **Olanzapine; 10 [507–516] | | | **Polyethylene glycol bowel preparation; 1 [517] |
| Trimethoprim-sulfamethoxazole; 9 [518–526] | Stibogluconate; 18 [527–535] | **Pantoprazole; 1 [536] | | | **Pregabalin; 1 [537] |
| **Vemurafenib; 1 [538] | Sulfasalazine; 8 [63,122,539–543] | **Propylthiouracil; 1 [544] | | | **Procetofene; 1 [58] |
| Valproic acid; 56 [545–582] | | **Riluzole; 3 [583–585] | | | **Rasburicase; 1 [586] |
| | **Valsartan; 1 [587] | **Rofecoxib; 2 [588,589] | | | Rifampin; 1 [590] |
| | **Voriconazole; 1 [591] | **Secnidazole; 1 [592] | | | Ritonavir; 1 [593] |
| | | **Sirolimus; 1 [594] | | | Roxithromycin; 1 [595] |
| | | **Theophylline; 1 [596] | | | **Stavudine; 1 [597] |
| | | **Tiaprofenic acid; 1 [598] | | | **Sunitinib; 1 [599] |
| | | **Tinidazole; 1 [600] | | | **Tacalcitol; 1 [601] |
| | | **Vedolizumab; 1 [602] | | | **Telmisartan; 1 [603] |
| | | **Vildagliptin; 2 [431,604] | | | **Tocilizumab; 1 [605] |
| | | | | | **Ursodeoxycholic acid; 1 [606] |
| | | | | | **Venlafaxine; 1 [607] |
| | | | | | **Zidovudine 1 [608] |
| | | | | | **Ziprasidone; 1 [609] |

Numbers of cases indicate the total cases identified for the drug; however, not all cases may have met the drug class criteria. Multiple cases may have been reported in a single reference, therefore, the number of citations may not equal the number of cases for each drug. See the supplement (**S5 Text**) for a detailed summary of each drug.
** New drug; association not reported by Badalov et al. [7].

Causality was assessed for 16 drugs (35% of Class Ib drugs) in 20 cases (11% of Class Ib cases. Three drugs demonstrated less than probable likelihoods of causation that were assessed in single case reports: clozapine, didanosine, and lamivudine were all assessed as possible. All other drugs for which causality was assessed demonstrated probable or definite likelihood in at least one case report.

Forty-one percent of the 175 cases (n = 72) in the Class Ib category had an underlying condition that may have predisposed them to AP. Twelve cases (7%) had underlying renal dysfunction that may have contributed to poor drug clearance and a predisposition to a drug reaction. Five cases had both an underlying condition and renal dysfunction potentially predisposing them to both AP and a drug reaction.

## Identified Class Ic drugs

Fifty-three drugs were classified as Class Ic from 106 case reports (15%), for a median of 1 case report per drug (mean = 2.0; mode = 1; range = 1–10). The four drugs reported in the highest number of cases were olanzapine (n = 10), clarithromycin (n = 5), liraglutide (n = 5), and minocycline (n = 5). Predominant latency categories, in order of frequency of occurrence amongst the 53 drugs, were 1–30 days (n = 31 drugs), >30 days (n = 19 drugs), and <24 hours (n = 10 drugs). The median latency consistency for the 26 drugs with multiple case reports was 71%.

Causality was assessed for 24 drugs (45% of Class Ic drugs) in 37 cases (35% of Class Ic cases. The following drugs demonstrated only possible likelihood of causation in at least one case report: clarithromycin, liraglutide, minocycline, nilotinib, and riluzole. All other drugs for which causality was assessed demonstrated probable or definite likelihood in all assessed case reports.

Thirty-five percent of the 94 cases (n = 33) in the Class Ic category had an underlying condition that may have predisposed them to AP. Thirteen cases (25%) had had a previous cholecystectomy. Ten cases (19%) had underlying renal dysfunction that may have contributed to poor drug clearance and a predisposition to a drug reaction. Four cases had both an underlying condition (diabetes) and renal dysfunction potentially predisposing them to both AP and a drug reaction.

## Identified Class II drugs

Six drugs were classified as Class II from 18 case reports (3%), including ceftriaxone (n = 5 cases), exenatide (n = 3), isotretinoin (n = 3), sitagliptin (n = 3), clofibrate (n = 2), and levetiracetam (n = 2). Three drugs had a predominant latency category of 1–30 days and three had a predominant latency cateogroy of >30 days. By definition for this drug class, all drugs had 100% latency consistency.

Causality was assessed for two drugs, exenatide and levetiracetam, both of which demonstrated probable likelihood of causation.

Ten of the 18 cases (56%) in the Class II category had at least one underlying condition that may have predisposed them to AP. Three cases (17%) had underlying renal dysfunction that may have contributed to poor drug clearance and a predisposition to a drug reaction. Two cases had both an underlying condition and renal dysfunction potentially predisposing it to both AP and a drug reaction. Four patients (22%) had had a cholecystectomy.

## Identified Class III drugs

Five drugs were classified as Class III from 13 case reports (2%), including gold (n = 4), acetylsalicylic acid (n = 3), nivolumab (n = 2), ondansetron (n = 2), and tacrolimus (n = 2). Latency consistency was 0% for the three drugs with two case reports each (i.e., both cases for each drug had different latencies); for the other two drugs, the median latency consistency was 71%.

Causality was assessed for one drug, tacrolimus, which demonstrated a probable likelihood of causation for one case.

The four cases with DIP due to gold had an immune disorder (all rheumatoid arthritis) that may have predisposed them to AP. No other cases had an underlying illness associated with AP or had reported risk factors for AP.

## Identified Class IV drugs

Fifty-seven drugs were classified as Class IV from single case reports (27% of all drugs; 8% of all cases). Latency was 1–30 days for 31 drugs, >30 days for 19 drugs, and <24 hours for 7 drugs. Re-challenge was conducted in one case report and was negative (brentuximab vedotin). Causality was assessed for 15 drugs (26% of Class IV drugs). Three drugs demonstrated less than probable likelihood of causation: alendronate, calcium carbonate, and ciprofibrate, all of which were assessed as possible. All other drugs for which causality was assessed demonstrated probable likelihood of causation.

Twenty-three of the 57 cases (40%) in the Class IV category had at least one underlying condition that may have predisposed them to AP. Two of these cases had multiple conditions associated with AP. Three cases (5%) had had a previous cholecystectomy. Five cases (9%) had underlying renal dysfunction that may have contributed to poor drug clearance and a predisposition to a drug reaction. Two cases had both an underlying condition and renal dysfunction potentially predisposing them to both AP and a drug reaction.

## Sensitivity analyses

When more stringent diagnostic criteria for AP were applied to the included cases (i.e., positive imaging), 429 of 713 cases (60%) met the restricted criteria, implicating 160 of 213 drugs (75%). The following numbers of drugs were found in each class:

- Class Ia: 31 drugs (from 45 drugs before imaging restriction)

- Class Ib: 33 drugs (from 46), four of which had been Class Ia drugs (all-trans-retinoic acid, cimetidine, interferon alpha, and thalidomide)

- Class Ic: 43 drugs (from 53), two of which had been Class Ia drugs (erythromycin and olsalazine) and two of which had been Class Ib drugs (amiodarone and paclitaxel)

- Class II: five drugs (from 6), one of which had been a Class Ib drug (stibogluconate)

- Class III: six drugs (from 5), two of which had been Class Ib drugs (carbamazepine and pentamidine)

- Class IV: 42 drugs (from 57), two of which had been Class Ia drugs (prednisone and sulindac), two of which had been Class Ib drugs (oxyphenbutazone and risperidone), three of which had been Class Ic drugs (bortezomib, metolazone, and naltrexone), and one of which had been a Class II drug (levetiracetam).

Fifty-three drugs were no longer associated with DIP, when cases were restricted to require positive imaging. Of these drugs, six were Class Ia (bezafibrate, metronidazole, orlistat, procainamide, ranitidine, and vemurafenib), eleven each were Class Ib and Class Ic, one each were Class II and III, and the remaining 23 were Class IV. More detailed lists of drugs from this sensitivity analysis can be found in the supplement (**S6 Text**).

When we required all ten causes of AP to be excluded instead of only four for a diagnosis of DIP, only two recently published cases met the more rigorous diagnostic criteria for DIP [14,15]. These two cases implicated amoxicillin/clavulanic acid and propofol, respectively. None of the remaining 711 cases met the more stringent DIP diagnostic criteria.

## Discussion

Several works have been published that have reviewed drugs associated with acute pancreatitis, both narratively and systematically [3–7], the most recent of which was published by Badalov et al. in 2007 [7]. Our initial goal was to update this review; however, early in the review we identified limitations in the reporting of the previous review's methods that prevented the conduct of a simple update. Consequently, we undertook a full systematic review of the case report literature, in its entirety, using rigorous systematic review methods. Our search strategy differed from that of Badalov et al. [7] in that we searched multiple databases, and ultimately included English, French, and Spanish publications. We used more strict screening criteria initially to capture only cases of AP based upon currently accepted diagnostic criteria [8], which corresponded to the "definite" and "probable" cases ultimately analyzed by Badalov et al [7]. However, we excluded cases in which a combination of drugs was the attributed cause of DIP, while Badalov et al. retained them. To synthesize the case reports, we adjusted the drug classification scheme proposed by Badalov et al., making all drug classes mutually exclusive. For classes II–IV, the revised scheme eliminated the classification criteria based upon arbitrary numbers of case reports published for a drug and focused instead on whether multiple (Classes II and III) or single (Class IV) case reports had been published. Our review identified substantially more drugs potentially associated with AP.

While a large number of drugs with potential associations with AP were identified in this review, these potential associations were based upon (a) cases that may not have had AP, and (b) cases of AP that had not completely excluded all non-drug causes of AP. All cases included in our review met currently accepted criteria for AP diagnosis; however, when more stringent diagnostic criteria were applied for the diagnosis of AP (i.e., positive imaging), the number of drugs associated with AP dropped by 25% (n = 160), suggesting that more rigorous evidence is required to confirm associations. Secondly, in our drug classification scheme, as partial assessment of causality, we followed the criteria proposed by Badalov et al. [7] and required exclusion of only four other causes of AP (i.e., alcohol, hypertriglyceridemia/hyperlipidemia, gallstones, and other drugs) rather than requiring exclusion of the remaining six non-drug causes proposed by our content experts (i.e., hypercalcemia, genetic/hereditary, anatomic, trauma, viral, and autoimmune causes). Had we required exclusion of all non-drug causes, only two drugs would have met the criteria of Classes Ia or Ic. In other words, the diagnosis of DIP was not conclusive in any of the 711 cases identified by authors as DIP in the literature. This sensitivity analysis demonstrates the tenuous level of evidence upon which DIP associations are often based and the contribution of comorbid conditions.

Awareness of all other possible non-drug causes of AP amongst clinicians may also influence formal causality assessment. The Naranjo algorithm, the most frequently encountered causality assessment tool in our set of included case reports, includes an assessment of whether all alternative etiologies of AP were excluded. However, if clinicians are unaware of all possible alternative causes of AP or the appropriate tests for their exclusion, then the final Naranjo score that determines the probability of causation may be inflated. Similarly, the probability of causation may be inflated when the prior probability of the event is not considered [611] (e.g., consideration of an inherent increased risk of AP in a patient due to factors such as the underlying disease for which the drug was administered, comorbidities, or concomitant drugs, and interactions of these factors). As well, valuable information regarding these other risk factors may be lost, if causation is distilled down to a simple probability level in causality assessment. Like a causality assessment, our drug classification system considers re-occurrence of AP upon re-challenge indicative of a higher level of evidence of an association. However, ethical concerns often prevent clinicians from knowingly reinitiating treatments suspected of causing

potentially life-threatening conditions like DIP. In our classification system, the absence of a positive re-challenge may occur for one of two reasons—no re-challenge was conducted or the presence of a negative re-challenge. However, our system did not penalize drugs that had a case with a negative re-challenge. We must recognize some of the limitations of causality assessment and not over-interpret the findings.

The complex relationships amongst underlying diseases, comorbidities, and concomitant drugs, as well as the difficulty in excluding all non-drug causes of AP and the constraints on re-challenge make final diagnosis of DIP extremely difficult. Case reports and the associations generated from their synthesis should be recognized as fallible. The evidence base upon which associations between drugs and AP are made could be bolstered through systematic review of observational studies or randomized controlled trials (RCTs) of individual drugs identified by case report review as having potential associations with AP. In RCTs, the control arm of patients with similar underlying diseases, comorbidities, and concomitant medications as those in the drug intervention arm reduces or eliminates the effects of these factors. Thus, the risk of AP due to the suspected drug is more easily discerned through the confounding factors and often is different from that gleaned from case report data. For example, an association between oral hypoglycemics and AP has been speculated based upon numerous published case reports [73–75,126,159,287–290,427–431,491,506,604]; however, systematic review and/or meta-analysis of trial data have demonstrated no association with AP for vildagliptin alone [612] or for dipeptidyl peptidase-4 inhibitors (DPP4i) as a group (i.e., vildagliptin, sitagliptin, saxagliptin, alogliptin, linagliptin, and dutogliptin) [613]. Another systematic review of various study designs concluded that AP during oral hypoglycemic therapy was a rare event and that if an association exists, it is not as strong as originally thought [614]. Similarly, meta-analyses of cohort and case-control study data have been recently published, refuting an association between statins and AP, of which several were implicated in cases included in our review [615]. Given recent efforts to evaluate associations between individual drug categories and AP, we considered categorizing the drugs implicated in our review by the AHFS system. However, ultimately, we consciously elected not to categorize and risk potentially promoting inferences about specific drug categories that were based upon low-quality evidence. Instead, we encourage the reader to explore the raw data set as well as the supplement available online.

There are strengths and limitations of the current review to be noted. In terms of strengths, we have ensured that the methods of our systematic review have been transparently reported to facilitate future updating within this area of active publication. We updated the study selection criteria of the previously published review [7] by including only those case reports that satisfied the Atlanta criteria for AP diagnosis [8]. However, we retained the previous review's requirement to exclude cases for which drug dosages were not reported, which may have restricted the volume of data available for synthesis to some degree. As well, while we excluded cases for which the implicated cause was a combination of drugs, in our included cases, it is possible that other drugs taken in proximity to or concurrently with the drug implicated by the authors could have made the patient more susceptible to the effects of the implicated drug. The same patient taking the same drug without exposure to another drug may not have experienced AP. As a strength, our drug classification system has eliminated some ambiguity in the previous system that was based upon the data extracted [7]. However, the classification system has some limitations. Firstly, we did not assess the testing methods used in case reports to exclude non-drug causes of AP because these data were often poorly reported. As such, older cases may have been misdiagnosed as DIP because the available testing wasn't sufficiently sensitive to identify some non-drug causes of AP, such as biliary sludge and crystal formation. Secondly, the classification system does not consider patients' underlying diseases. Some conditions for which drugs are administered may predispose patients to develop AP (e.g., HIV/

AIDS [5]) or be susceptible to drug-induced adverse events (e.g., renal dysfunction). Almost half of all cases included in this review had an underlying disease that may have predisposed them to AP. We have presented data regarding the presence of underlying conditions to guide readers. Finally, drugs implicated in multiple case reports were classified based upon the case report meeting the highest criteria for classification, rather than the median class for the group. With this method of classification, drugs can never be demoted as further case reports are published, a potential limitation for updating the review. Future updates may need to consider data from study designs other than case reports (e.g., meta-analyses of adverse event data from trials) for drugs with demonstrated ambiguous associations (e.g., oral hypoglycemics). However, it should be noted that while meta-analysis of adverse event data from RCTs may help to discern the relationship between a drug and DIP, these analyses are often limited by the complexity of diagnosis of AP and the lack of detailed reporting of adverse events within these trials. Finally, while this review incorporated extensive efforts to find a large amount of information to be dissected to meet our objectives, it must be kept in mind that the type of evidence reviewed (i.e., case reports) remains at the bottom of the hierarchy of evidence and, thus, has inherent limitations for inferences.

## Conclusions

Although the rate of publication of case reports has lowered in the past decade, we continue to identify drugs with new associations with AP and new evidence to bolster or refute previously suspected associations. Much of the case report evidence upon which DIP associations are based is tenuous. A greater emphasis on exclusion of all non-drug causes of AP and on quality reporting would improve the evidence base considerably. However, even with improvements in methods and reporting, it should be recognized that case reports remain the lowest level in the hierarchy of evidence [616], and that case report reviews, although valuable scoping tools of the published post-market evidence, should be recognized as having limited strength in the establishment of associations between drugs and AP.

## Supporting information

**S1 Text. Literature search strategy.**
(DOCX)

**S2 Text. Completed PRISMA checklist.**
(DOCX)

**S3 Text. List of included studies.**
(DOCX)

**S4 Text. List of excluded studies.**
(DOCX)

**S5 Text. Detailed summary table–drugs associated with drug induced pancreatitis.**
(DOCX)

**S6 Text. Sensitivity analysis–positive imaging findings required for a diagnosis of DIP.**
(DOCX)

**S1 Data. Collection of extracted data.**
(XLSX)

## Acknowledgments

We wish to thank Raymond Daniel for his indispensable support in reference acquisition and database management throughout the conduct of this review.

## Author Contributions

**Conceptualization:** Salmaan Kanji, Brian Hutton.

**Data curation:** Dianna Wolfe, Fatemeh Yazdi, Pauline Barbeau, Danielle Rice, Andrew Beck, Claire Butler, Leila Esmaeilisaraji.

**Formal analysis:** Dianna Wolfe, Brian Hutton.

**Funding acquisition:** David Moher, Brian Hutton.

**Methodology:** Dianna Wolfe, Salmaan Kanji, Danielle Rice, Becky Skidmore, David Moher, Brian Hutton.

**Project administration:** Brian Hutton.

**Resources:** Fatemeh Yazdi, Pauline Barbeau, Danielle Rice, Claire Butler, Leila Esmaeilisaraji, Becky Skidmore, Brian Hutton.

**Software:** Becky Skidmore.

**Supervision:** Salmaan Kanji, Brian Hutton.

**Validation:** Fatemeh Yazdi, Pauline Barbeau, Andrew Beck, Claire Butler, Leila Esmaeilisaraji, Brian Hutton.

**Visualization:** Dianna Wolfe.

**Writing – original draft:** Dianna Wolfe, Brian Hutton.

**Writing – review & editing:** Dianna Wolfe, Salmaan Kanji, Fatemeh Yazdi, Pauline Barbeau, Danielle Rice, Andrew Beck, Claire Butler, Leila Esmaeilisaraji, Becky Skidmore, David Moher, Brian Hutton.

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
