## [Decision Letter · Decision Letter 0]

4 Mar 2020

PONE-D-20-02739

Drug Induced Pancreatitis: a systematic review of case reports to determine potential drug associations

PLOS ONE

Dear Dr. Hutton,

Thank you for submitting your manuscript to PLOS ONE. After careful consideration, we feel that it has merit but does not fully meet PLOS ONE’s publication criteria as it currently stands. Therefore, we invite you to submit a revised version of the manuscript that addresses the points raised during the review process.

We would appreciate receiving your revised manuscript within 3 months. To enhance the reproducibility of your results, we recommend that if applicable you deposit your laboratory protocols in protocols.io, where a protocol can be assigned its own identifier (DOI) such that it can be cited independently in the future. For instructions see: http://journals.plos.org/plosone/s/submission-guidelines#loc-laboratory-protocols

We look forward to receiving your revised manuscript.

Kind regards,

Francisco X. Real

Academic Editor

PLOS ONE

Journal Requirements:

Reviewers' comments:

Reviewer's Responses to Questions

**Comments to the Author**

1. Is the manuscript technically sound, and do the data support the conclusions?

Reviewer #1: Yes

Reviewer #2: Yes

2. Has the statistical analysis been performed appropriately and rigorously? 

Reviewer #1: Yes

Reviewer #2: Yes

3. Have the authors made all data underlying the findings in their manuscript fully available?

Reviewer #1: Yes

Reviewer #2: Yes

4. Is the manuscript presented in an intelligible fashion and written in standard English?

Reviewer #1: Yes

Reviewer #2: Yes

5. Review Comments to the Author

Reviewer #1: This is a very interesting review dealing with the topic of drug-induced AP. The authors carefuly defined criteria to revise the available literature and conclude that evidence is typically very low in most instances for the reported assoications.

The authors should be congratulated for the big effort and for the clarity of the presentation.

I do have minor comments:

1. While I agree that including only studies that accomplish Atlanta criteria is ok, I would ask to make a further sensitivity analsysis on studies that had positive CT scan or MRI. The other two crietria (pain and elevated levels of pancreatic enzymes) can occurr in digestive disorders (typically i nIBD) without having an AP. Please try and do that and comment.

2. Among the drugs, especially in class I and II there are a lot of statins. A recent systematic review and meta-analysis (PMID: 30288283 )on C-C and cohort studies concluded that there is no evidence for such drugs causing AP. this should be commented and discussed.

Reviewer #2: The manuscript by Hutton is a tour de force on the topic of case reports on drug associated acute pancreatitis. The authors are to be commended on going back to the primary sources. The manuscript is primarily descriptive, with extracted data being organized and classified into tables. This largely verifies what is known - i.e. the source material in clinical reports is poor.

The manuscript would be strengthened with some attempts to classify the drugs by mechanistic classes. For example, are they channel blockers, anticholinergic agents, specific types of antibiotics, etc. (They did address hyperglycemic agents in the discussion as a class). Otherwise, it is a lot of data that is hard to synthesize and use.

6. PLOS authors have the option to publish the peer review history of their article (what does this mean?). If published, this will include your full peer review and any attached files.

Reviewer #1: No

Reviewer #2: Yes: David C. Whitcomb MD PhD

---

## [Author Response · Author response to Decision Letter 0]

14 Mar 2020

March 14, 2020

Dear Editor:

On behalf of my co-authors, I would like to re-submit our revised manuscript “Drug Induced Pancreatitis: a systematic review of case reports to determine potential drug associations”. We thank you for the opportunity to revise and resubmit this manuscript. We appreciate the favorable review of our manuscript and we thank the reviewers for the helpful comments provided. 

We have responded to each reviewer comment and have revised the manuscript to reflect these responses. Point-by-point responses to all comments are noted below, as are notations of the places in the text where changes have been made to address the reviewers’ suggestions. We have submitted a “Track Changes” and “Clean – No Track Changes” version of the manuscript. 

Please note that in error, one co-author was accidentally omitted from the initial submission (Danielle Rice); she has been appropriately added for the re-submission.

We hope that you agree that the revised manuscript has addressed the concerns raised by the reviewers and is now ready for publication in PLOS One.

Sincerely, 

Brian Hutton, MSc, PhD 

Director and Senior Scientist | Knowledge Synthesis Group | Clinical Epidemiology Program

Assistant Professor, Ottawa University School of Epidemiology and Public Health

Adjunct Scientist, Royal Ottawa Institute of Mental Health Research

Ottawa Hospital Research Institute

Center for Practice Changing Research Building

The Ottawa Hospital - General Campus

501 Smyth Road | PO Box 201B | Ottawa | Ontario | Canada | K1H 8L6

email: bhutton@ohri.ca

Phone: 613-737-8899 (ext. 73842) | Fax: 613 - 739 - 6938

Reviewer Comments

Reviewer #1: 

This is a very interesting review dealing with the topic of drug-induced AP. The authors carefully defined criteria to revise the available literature and conclude that evidence is typically very low in most instances for the reported associations. The authors should be congratulated for the big effort and for the clarity of the presentation. I do have minor comments:

1. While I agree that including only studies that accomplish Atlanta criteria is ok, I would ask to make a further sensitivity analsysis on studies that had positive CT scan or MRI. The other two criteria (pain and elevated levels of pancreatic enzymes) can occur in digestive disorders (typically in IBD) without having an AP. Please try and do that and comment.

2. Among the drugs, especially in class I and II there are a lot of statins. A recent systematic review and meta-analysis (PMID: 30288283) on C-C and cohort studies concluded that there is no evidence for such drugs causing AP. This should be commented and discussed.

Response: We thank Reviewer 1 for this high praise, it was indeed a considerable amount of work. To augment the submitted findings, we have conducted a brief sensitivity analysis, as proposed by the reviewer, and revised the manuscript accordingly. An additional appendix has been added to the supplement, with more detailed findings from the sensitivity analysis (see the updated Appendix S6 Text); we invite the reviewer to inspect the tracked changes version of the re-submitted manuscript to see all of the text additions made to allude to findings from the sensitivity analysis. We have also added a line regarding the suggested systematic review in the Discussion section which reads as follows: “Similarly, meta-analyses of cohort and case-control study data have been recently published, refuting an association between statins and AP, of which several were implicated in cases included in our review.”

Reviewer #2: 

The manuscript by Hutton is a tour de force on the topic of case reports on drug associated acute pancreatitis. The authors are to be commended on going back to the primary sources. The manuscript is primarily descriptive, with extracted data being organized and classified into tables. This largely verifies what is known - i.e. the source material in clinical reports is poor.

The manuscript would be strengthened with some attempts to classify the drugs by mechanistic classes. For example, are they channel blockers, anticholinergic agents, specific types of antibiotics, etc. (They did address hyperglycemic agents in the discussion as a class). Otherwise, it is a lot of data that is hard to synthesize and use.

Response: We thank Reviewer 2 for the glowing comments. We considered categorizing the implicated drugs by the AHFS system; however, subsequently we recognized that the level of evidence that we were synthesizing was low, and we then consciously elected not to promote inferences about drug categories based upon what many would likely consider to be very weak evidence. We have added a comment in the Discussion section regarding this consideration, which reads as follows: “Given recent efforts to evaluate associations between individual drug categories and AP, we considered categorizing the drugs implicated in our review by the AHFS system. However, ultimately, we consciously elected not to categorize and risk potentially promoting inferences about specific drug categories that were based upon low-quality evidence. Instead, we encourage the reader to explore the raw data set as well as the supplement available online.” 

We have provided the original file of raw data online, and would encourage readers to develop their own inferences, keeping in mind the low level of evidence.

---

## [Decision Letter · Decision Letter 1]

3 Apr 2020

Drug Induced Pancreatitis: a systematic review of case reports to determine potential drug associations

PONE-D-20-02739R1

Dear Dr. Hutton,

We are pleased to inform you that your manuscript has been judged scientifically suitable for publication and will be formally accepted for publication once it complies with all outstanding technical requirements.

With kind regards,

Francisco X. Real

Academic Editor

PLOS ONE

Additional Editor Comments (optional):

Reviewers' comments:

Reviewer's Responses to Questions

**Comments to the Author**

1. If the authors have adequately addressed your comments raised in a previous round of review and you feel that this manuscript is now acceptable for publication, you may indicate that here to bypass the “Comments to the Author” section, enter your conflict of interest statement in the “Confidential to Editor” section, and submit your "Accept" recommendation.

Reviewer #1: All comments have been addressed

2. Is the manuscript technically sound, and do the data support the conclusions?

Reviewer #1: Yes

3. Has the statistical analysis been performed appropriately and rigorously? 

Reviewer #1: Yes

4. Have the authors made all data underlying the findings in their manuscript fully available?

Reviewer #1: Yes

5. Is the manuscript presented in an intelligible fashion and written in standard English?

Reviewer #1: Yes

6. Review Comments to the Author

Reviewer #1: The present version of the manuscript is acceptable for publication after the requested changes have been made. I do not have additional comments.

7. PLOS authors have the option to publish the peer review history of their article (what does this mean?). If published, this will include your full peer review and any attached files.

Reviewer #1: No

---

## [Editor Report · Acceptance letter]

6 Apr 2020

PONE-D-20-02739R1 

Drug Induced Pancreatitis: a systematic review of case reports to determine potential drug associations 

Dear Dr. Hutton:

I am pleased to inform you that your manuscript has been deemed suitable for publication in PLOS ONE. Congratulations! Your manuscript is now with our production department. 

With kind regards,

on behalf of

Dr. Francisco X. Real 

Academic Editor

PLOS ONE